# Negative and Positive Predictors of Anastomotic Leakage in Colorectal Cancer Patients—The Case of Neutrophil-to-Lymphocyte Ratio

**DOI:** 10.3390/diagnostics14161806

**Published:** 2024-08-19

**Authors:** Aristeidis Ioannidis, Georgios Tzikos, Aikaterini Smprini, Alexandra-Eleftheria Menni, Anne Shrewsbury, George Stavrou, Daniel Paramythiotis, Antonios Michalopoulos, Katerina Kotzampassi

**Affiliations:** 1Department of Surgery, Aristotle University of Thessaloniki, 54636 Thessaloniki, Greece; ariioann@yahoo.gr (A.I.); tzikos_giorgos@outlook.com (G.T.); ksmprini@hotmail.com (A.S.); alexmenn@auth.gr (A.-E.M.); a_shrewsbury@yahoo.com (A.S.);; 2Department of Colorectal Surgery, Addenbrooke’s Hospital, Cambridge CB2 2QQ, UK; stavgd@gmail.com

**Keywords:** neutrophil-to-lymphocyte ratio (NLR), colorectal cancer, anastomotic leakage, anastomotic leak, anastomotic dehiscence, prognostic factor, lymphocytopenia

## Abstract

Colorectal surgery for cancer is associated with a high rate of surgical complications, including anastomotic leakage. The ability to predict the risk of leakage early enough seems to be of high value, since it would facilitate the design of personalized treatment and duration of hospitalization. Although different studies present the neutrophil-to-lymphocyte ratio [NLR] as having a strong predictive value, there is a discrepancy with respect to which postoperative day is the most reliable. We evaluated a series of NLR values, from the day before surgery up to the POD7, in a cohort of 245 colorectal surgery patients in order to clarify the best predictable score for the identification of the risk of anastomotic leakage. There were 28 patients with leaks. ROC curve analysis of NLR on POD1 indicates that a cut-off point ≥ 7.4 exerts a negative prediction for leakage (AUC 0.881, sensitivity 68.7%, specificity 96.4%, PPV 28.4%, and NPV of 99.3%), thus excluding 150 patients from the risk of leakage. Furthermore, the ROC curve analysis of NLR on POD4 indicates that a cut-off point ≥ 6.5 gives a positive prediction of leakage (AUC 0.698, sensitivity 82.1%, specificity 51.6%, PPV 17.6%, and NPV of 95.6%), thus indicating 52 patients as being at high risk of leakage. Finally, NLR failed to identify five leaks out of twenty-eight. These results strongly indicate the ability of NLR on POD1 to predict patients at low risk of developing a leak and then on POD4 to predict the high-risk patients. This makes our study particularly innovative, in that it enables doctors to concentrate on potential high-risk patients from POD1.

## 1. Introduction

Despite advances in anesthesia, surgical techniques, and perioperative care, including nutritional support, anastomotic leakage continues to be a serious complication—all too often an unfortunate reality—and a significant source of harm for both patients and surgeons. Specifically, in the cases of colorectal surgery, in which improvements in minimally invasive technology and preoperative neoadjuvant therapy open the way for low or even ultra-low anus-preserving resections, an anastomotic leakage, or simply a dehiscence, considerably increases postoperative morbidity, risk for a permanent stoma, and hospitalization length and cost, while it delays adjuvant therapy, finally worsening the oncologic outcomes, or leads to death [1,2,3].

Anastomotic leakage often occurs between the 5th and 8th postoperative day [4], regardless of the anatomical area and the intensity of clinical manifestation; however, in the period of enhanced recovery after surgery, approximately 20% of leaks are usually recognized after patient discharge from hospital [3,5]. Thus, the identification of a reliable, biological, prognostic marker—preferably easy to process and of low cost—remains a high priority for colorectal surgery, in order to facilitate an early diagnosis and a timely intervention.

For some years, the neutrophil-to-lymphocyte ratio (NLR) has been used as a simple marker of subclinical inflammatory response, applicable to a variety of disease conditions, from carcinomas to traumas and from abdominal surgery to cardiovascular diseases, reflecting the immune system’s sufficiency and its ability to “fight” inflammation. It exhibits a positive correlation to disease severity, progress, morbidity and mortality, and thus it has also been tried as a predictor of major postoperative complications in abdominal surgery [3,6,7,8,9,10]. Additionally, it has been used as an indicator of colorectal tumor staging [11] as well as a predictor of poor long-term postoperative prognosis [12]. However, its predictive role regarding the early recognition of anastomotic leakage in colorectal cancer patients is still unclear and needs to be further evaluated. Almost all studies are in agreement that NLR should be used as a reliable prognostic index of high sensitivity and specificity, but researchers consider different postoperative days as the most likely, from the day before surgery up to the 6th postoperative day, the different findings probably related to the dissimilar population of each study and the diversity of individual patients characteristics, such as their immunological and nutritional status, their discrete gut microbiota diversity, their age and the invasiveness of the disease, the preoperative neo-adjuvant therapy or not [13,14,15,16], the ratio of low and very low anterior resections in the sample size, the construction of a prophylactic diverting stoma, and even the precise definition of anastomotic leakage used in the study [17,18,19], all of which are probably involved in this confusion. 

Thus, in order to lend clarity to this question, we conducted a new study aiming to calculate the daily NLR value from the preoperative day (POD0) to the 7th postoperative day (POD7) in order to assess the best predictive value as a marker of anastomotic leakage in a group of patients, half of whom had been subjected to low anterior resection.

## 2. Materials and Methods

### 2.1. Patients

This is a retrospective, single-center analysis of data retrieved from the electronic files of all patients electively operated for colorectal cancer between January 2018 and December 2023. Patients, to be included in the study, had to have been subjected—during the same procedure—to an elective colonic resection of any type, along with a primary end-to-end anastomosis, and, optionally, a preventive colostomy or ileostomy, at the discretion of the colorectal surgeon, when the resection involved the rectum.

Patients subjected to emergency operations, those considered as non-operable at the time of surgery, or those subjected to an elective resection not competed by an anastomosis were automatically excluded from the study. Additionally, patients having undergone laparoscopic colectomy or reoperated for any reason, or who had no complete hematological data in their electronic file, were also excluded (Figure 1).

The patients finally included had been operated on—after mechanical bowel preparation plus a dose of prophylactic antibiotics at induction—by the same experienced colorectal specialist in various teams. Nobody received supplementary epidural anesthesia; a circular stapler was used for low anterior resections and a pelvic drainage tube was also routinely placed; for all the other anastomoses, a linear stapler was used. All patients received the same standard care, according to the usual practice of the responsible consultant surgeon.

### 2.2. Definition of Anastomosis Dehiscence/Leakage

The diagnosis of anastomotic dehiscence/leakage was based on the standard definition proposed by the International Study Group of Rectal Cancer (ISREC) in 2010; a leak is “a defect of the intestinal wall at the anastomotic site, leading to a communication between the intra- and extraluminal compartments” [20]. Further, the severity of anastomotic leakage was graded according to the impact on clinical management: Grade A, being asymptomatic, resulted in no change in patient management; Grade B required active therapeutic intervention, but was manageable without re-laparotomy; and Grade C required re-laparotomy [19,20,21].

### 2.3. Data Collection

Initially, data from all patients meeting all the above criteria and thus included in the study were collected: age, gender, and type of operation, and neutrophil and lymphocyte counts—in absolute numbers and not as percentages—on days 0 (POD0, preoperatively) and on days 1 (POD1) to 7 (POD7) postoperatively. These data were anonymously tabulated, the only reference being the patients’ hospital admission number. The NLR ratio on every day (8 measurements) was then calculated for each patient/day by dividing the absolute neutrophil count by the absolute lymphocyte count.

In parallel, another blinded investigator collected all the cases of anastomotic leakage reported in patients’ discharge forms, along with their hospital admission number, in order to cross-match them to the initial data collected.

Finally, the cases of an anastomotic dehiscence—unexpectedly recognized during endoscopy performed before ileostomy/colostomy closure by the end of the 3rd postoperative month—were also collected and matched according to the admission number which the patient had received at the time of colectomy. This search was focused only on patients subjected to a prophylactic ileostomy or colostomy at the time of elective low anterior resection plus anastomosis, and did not include any patient who had a salvage stoma performed in an emergency re-operation. These patients were considered as having experienced a leak of Grade A—asymptomatic. 

Based on the unique criterion of intact or ruptured anastomosis, we separated the patients into two main groups, the non-leak (non-L) and the leak (L) groups. The L group was further subdivided into those patients who experienced a clinically and/or CT imaging documented leakage—that is, of Grade B and C—and those being asymptomatic, Grade A, retrospectively recognized at endoscopy.

### 2.4. Statistical Analysis

Statistical analysis was conducted with the Statistical Package for Social Science (SPSS), Inc. (v 25.0; Chicago, IL, USA). The data’s compliance with normal distribution was assessed by the Kolmogorov–Smirnov or Shapiro–Wilk test when the evaluated group included more or fewer than 50 patients, respectively. For continuous variables, the results were presented as mean ± standard deviation (SD), normality being assumed, and as median and interquartile range (IQR) for variables with skewed distribution. Moreover, to compare the mean values of the two independent groups, Student’s sample t test was used for parametric data, and the Mann–Whitney U test for nonparametric data. Subsequently, a Wilcoxon Rank test was performed to evaluate the differences in the means of two interdependent samples when comparing the values within the same group, but at different times. Qualitive data were presented as percentages and the chi-square test was used for the analysis of the nominal variables. Moreover, a graphical presentation of neutrophil and lymphocyte values, as well as of NLRs, in the two groups, and their statistical comparison per day was made. 

Finally, a receiver operation characteristic (ROC) curve was generated to assess the cut-off points with the best diagnostic performance regarding anastomotic leakage. The optimal cut-off values were determined according to the Youden index [22,23]. Sensitivity, specificity, Positive Predictive Value (PPV), Negative Predictive Value (NPV), and Positive Likelihood Ratio were calculated as well. The statistical significance level was set at 0.05.

## 3. Results

### 3.1. Patients

Two hundred and forty-five patients fulfilling the inclusion criteria were finally included in the analysis. Eighteen patients (rate: 7.34%) experienced an anastomotic leakage—12 of Grade B and 6 of Grade C—requiring re-operation. Another 10 patients were added to the L group, although they would have remained undiagnosed if there had not been, by protocol, an endoscopic inspection of the anastomotic suture line before ostomy closure by the end of the 3rd postoperative month. These very low anterior resection patients were found to have a suture line disruption of less than a quarter of the circle, and, since being totally asymptomatic, they were classified as Grade A. The remaining 217 patients had an uneventful postoperative period, thus comprising the non-L group. 

Demographic characteristics and the clinical data of patients with and without anastomotic leak are summarized in Table 1.

### 3.2. Neutrophil Counts

Daily neutrophil counts (normal values 1.6–6.5 × 10^9^/L) were found to have no significant difference between the two groups on the day before surgery (POD0) as well as on all seven postoperative days, except for POD2; on this day, the L group’s neutrophil count exhibited only a small increase, in relation to the non-L group, leading to a highly significant difference (*p* < 0.001) between the two groups (Table 2). As can be seen in Figure 1, neutrophils of the non-L group presented a sharp increase from POD0 up to POD1, the value flattening on POD2 and then declining smoothly, and returning to almost the POD0 value. 

However, neutrophils of the L group presented a smooth increase from POD0 to POD1 and continued increasing up to POD5 to reach the value of POD1 of the non-L group, then smoothly declined. These differences in curve lines of both groups led to the maximum distance of values on POD1 (*p* < 0.001). 

### 3.3. Lymphocyte Counts

Daily lymphocyte counts (normal values: 1.0–4.8 × 10^9^/L) were found to have no significant difference between the two groups on POD0, POD1, and POD2, as well as on POD6 and POD7. On POD2 and thereafter, and only in the L group, was there a significant drop in lymphocyte count, to below the normal values—lymphopenia—up to POD6, when the values normalized again; this is why there is a statistically significant difference in mean values between the two groups on POD3, POD4, and POD5 (Table 3). As can be seen in Figure 2, lymphocytes of non-L group exhibited a sharp drop on POD1, as expected, due to operative stress and then slowly recovered to the initial, normal values. On the contrary, in L group a progressive drop continued up to POD5 (the minimum value on POD3), when a smooth recovery began.

### 3.4. NLR Counts

Daily mean values of the NLR, as the ratio of neutrophils to lymphocytes, are presented in Table 4. As can be seen in Figure 3, the POD0 values were quite similar; the same occurring with the POD2 values. However, on POD1, there was a highly significant difference, due to the very low NLR value of the L group (*p* < 0.001); then from POD3 thereafter, there was a highly significant difference, this time due to the drop in daily values in the non-L group, as a result of lymphopenia. In other words, the major distances between value-points on the same day was on POD1, where the L group had a lower value, and then on POD3, POD4, and POD5, where the L group had a higher value; this indicates the time points where there is a clear indication of prediction of a leakage. 

### 3.5. The Receiver Operating Characteristic Curve (ROC) Analysis

Since there was a statistical difference between the NLR values in both groups on six out of the eight days of NLR calculation, we performed a ROC analysis daily for the NLR values in order to determine the optimal cut-off values for predicting postoperative complications. This ROC analysis revealed that the area under the curve (AUC) was statistically significant for the NLR values of POD1 and from POD3 and thereafter (Table 5), while the diagnostic accuracy of NLRs is presented in Table 6.

Based on the above results, we are able to conclude the following: ROC curve analysis of the NLR on POD1 indicates that a cut-off point ≥ 7.4 exerts a negative prediction for leakage, with an AUC of 0.881, a sensitivity of 68.7%, a specificity of 96.4%, a Positive Predictive Value (PPV) of 28.4%, and a Negative Predictive Value (NPV) of 99.3%. The Positive Likelihood Ratio for not exhibiting a leak was calculated as 3.08 (2.50–3.79) (Table 7, Figure 4).

This finding means practically that, on POD1, from the total pool of 245 patients, 150 were found to have an NLR value ≥ 7.4; thus, this subgroup can be characterized as having a low probability of exhibiting an anastomotic leak (due to high specificity of the NLR on POD1 to predict the event of no leak). Indeed, in this material, 149 out of these 150 patients (99.3%) remained free of leakage. Subsequently, NLRs provided the information which helped to safely discriminate low-risk patients. 

ROC analysis of the NLR on POD3 gives a good prediction of leakage with an optimal cut-off point of 11.90; with an AUC of 0.632, a sensitivity of 50.0%, a specificity of 76.0%, a PPV of 21.2%, and a NPV of 92.2%. Furthermore, the ROC curve analysis of the NLR on POD4 indicates that a cut-off point ≥ 6.5 gives a better positive prediction for leakage (AUC of 0.698, a sensitivity of 82.1%, a specificity of 51.6%, a PPV of 17.6%, and a NPV of 95.6%) (Figure 5).

On POD1, we had excluded from the initial pool of 245 patients, 150 as considered of low risk; of the remaining 95 patients, 43 presented with an NLR < 6.50 and, based on the performance of the NLR on POD4, these 43 could theoretically be considered as a low probability for presenting a leakage. Consequently, only four out of forty-three patients were included as false negative results, which means that—although they had an NLR < 6.50—they did exhibit a leakage. 

Taking the above into consideration, in the sample of 245 patients and based on POD1 NLR screening (after exclusion of the 150), we were able to focus on fewer than half of the patients (95 out of 245 patients, 38.8%) who were considered high risk for a leak. Thus, on POD4, we could identify the low-risk patients, who accounted for 45.2% (43 out of 95) of this subgroup. So already, from POD4 onwards, we needed to be alert for anastomosis leakage only for 21.2% (52 out of 245) of the patients of the initial sample. However, NLRs failed to identify five leaks (one on POD1 and four on POD4), of which one patient was Grade C (re-operation), two were Grade B, and two were Grade C (Figure 6).

## 4. Discussion

Today, the neutrophil-to-lymphocyte ratio (NLR) is one of the most accepted prognostic indices in diseases of different etiology, generally reflecting the sufficiency of an individual’s immune system and its ability to “fight” inflammation [6,24]. It seems to predict the development of a systemic inflammation, and its deterioration or improvement, as a better indicator than the leukocyte, neutrophil, or lymphocyte count alone [25,26].

In the case of patients subjected to a surgical operation, besides the “disease” itself and the psychosomatic stress, the impact of anesthesia and surgical manipulations, along with the postoperative pain and other iatrogenic parameters (malnutrition, preoperative feeding, inotropes, gut dysmotility, etc.) lead to a strong neuroendocrine system response, part of the consequences of which is the increase in circulating neutrophils, accompanied by a fall in circulating lymphocyte levels [27,28,29]. In other words, the NLR values of the early postoperative days are strong associated with the postoperative stress response [30], which reflects the body’s inflammatory response, as well as its potential for self-repair.

Colorectal surgery, despite recent advances in patients’ preoperative support, the evolution of minimally invasive surgical techniques, mechanically improved stapling devices, and overall postoperative management with early rehabilitation and reduced hospital stay, continues to be associated with high morbidity rates [3,31,32]. Anastomotic leakage is one of the most severe and even life-threatening complications, reported to occur in almost 15% or more of, and up to 27% of, patients, while accounting for up to one third of the overall mortality in these procedures [7,33,34].

Anastomotic leakage frequently occurs between the 5th and 8th postoperative days, independently of the severity of its clinical manifestation; however, in the era of Fast Track surgery, approximately 20% of ruptures are usually recognized after a mean of 6 to 15 days after patient discharge [4,5,35]. Thus, the establishment of a low-cost and reliable, biological, prognostication marker, as is the NLR, remains of high priority for colorectal surgery, since it leads to a decrease in morbidity and mortality—by means of alerting the surgeons—and reduces the length of in-hospital stay, likelihood of re-admission, and overall costs, not to mention the poorer oncological outcomes for the patients [4,7,36,37,38,39].

To the best of our knowledge, there are ten clinical studies positively indicating the predictive role of NLRs in anastomotic leakage; however, there is a severe discrepancy as to which day is the most reliable. As early as 2007, Cook et al. [40], in a small group (n = 100) of elective colorectal resection patients, suggested that surgical manipulation affects NLR value from POD1; patients who went on to experience complications had a higher NLR, with a cut-off value greater than 9.3 (AUC 0.66, with a sensitivity of 66% and a specificity of 69%). A further three studies support the option of the preoperative NLR value as the most predictive: Caputo et al. [41] found that patients subjected to neo-adjuvant radio-chemotherapy before rectal surgery and having a preoperative NLR score above the cut-off 3.8 had significantly higher rates of postoperative complications. Josse et al. [42], on 583 colorectal cancer patients, reported that a preoperative NLR ≥ 2.3 was significantly associated with a major perioperative complication, with no relationship being found between the increased NLR and the type of complication, although there was a trend towards the occurrence of anastomotic leakage. One year later, Miyakita et al. [43], on 260 rectal cancer patients (56 complications), showed that a preoperative NLR cut-off point ≥ 2.21 was significantly associated with postoperative leakage (*p* = 0.0033), having a sensitivity of 83% and a specificity of 47%. In a large, multicenter study (1432 patients), Paliogiannis et al. [4] demonstrated that the NLR on both POD1 and POD4 was significantly higher (0.007 and 0.0001, respectively) in patients who finally experienced a leak; however, on POD4 NLR cut-off score of 7.1 showed the best prediction of a leakage (AUC 0.744, 72.7% sensitivity, and specificity 73.4%). Similarly, in a high-volume cohort of 1328 colon surgery patients, Benlice C. et al. [44] found a NLR > 9.2 to be a significant predictor of complications, of any kind, on POD2 (OR = 1.43; 95% CI: 1.03–1.98; *p* = 0.02), while similar results were reported by Mik et al. [7] on POD4 in 724 patients—33 leaks (4.6%). NLRs had a statistically significant difference (*p* = 0.0012) between those experiencing a leak (9.03 ± 4.13) and no-leaks (4.45 ± 2.25), with an AUC of 0.68 at a cut-off point of 6.5 (69% sensitivity, 78% specificity, positive predictive value of 49%, and negative predictive value at 88%). Additionally, findings with 136 patients—11 leaks (rate 8.1%)—also support the use of the NLR score calculated on POD4 (AUC 0.78, cut-off value 6.15, with a sensitivity of 100% and a specificity of 61.8%) as a useful predictor of a leak [45]. Finally, Shelygin YA, et al. [46] found a cut-off point of 5.13 (AUC 0.644, sensitivity 69.7%, and specificity 60.7%, *p* = 0.019) on POD3 and a cut-off point of 3.94 (AUC 0.75, sensitivity 75.9%, and specificity 70.6%, *p* < 0.001) on POD6 to be of predictive value in a cohort of 192 patients, and Tan F, et al. [3] a cut-off score of 6.54 (AUC 0.818, *p* < 0.001, with a sensitivity of 76.5% and a specificity of 79.4%) on POD5, in 306 patients—17 (5.56%) with symptomatic leaks.

The above analysis clearly underlines the problem: all researchers are in agreement as to the reliability of the NLR as a prognostic index, but each author considers a different postoperative day as the most predictive, from the day before surgery up to POD6. The reasons for this discrepancy are fairly obvious, since the population of each study is quite dissimilar: Patients differ in respect to the immunological and nutritional status, their gut microbiota diversity, age and gender, tumor invasiveness, and whether or not they have been subjected to preoperative neo-adjuvant therapy. Additionally, the ratio of low and very low anterior resections in the sample size, the sample size itself, the construction of a prophylactic diverting stoma, and certainly the precise definition of anastomotic leakage used by the authors all likely to contribute to this confusion. Our study seems to include the maximum number of low anterior resections, 122 patients against 123 with cancer in other locations, and it is well known that the more distal the anastomosis, the higher the risk of leakage, with low colorectal and coloanal anastomoses having the highest leak rate. Additionally, the detrimental effects of preoperative neoadjuvant therapy on wound healing are likely to results in increased leakage [47]. Finally, almost none of the papers referring to the NLR index define what exactly they mean by “anastomotic leakage”. Bruce et al., in a review of 97 studies, referred to anastomotic leaks, counted 29 different definitions for lower GI anastomotic leaks [48]. Thus, the lack of a uniform definition makes the true incidence unknown and comparisons between studies difficult [17], since some studies include only leaks that required some mode of treatment, and others include all cases of apparent leak on imaging, regardless of clinical course [21], as occurred in our material.

Our findings reveal that patients having a NLR cut-off equal or greater than 7.4 on POD1 are safe. This was true for 149 out of the 150 patients having such a score, easily translated into those patients having a good immunological status, able to control and thus avoid an excessive inflammatory response. Additionally, we report a positive predictive value on POD4: individuals having a NLR cut-off ≤ 6.5 being of low risk of an anastomotic leak—this being the case for 39 out of 43 patients with such a score—while the rest from the 95 (n = 52), with a cut-off ≥ 6.5, were considered to be of high risk. In other words, on POD4, we could identify the low-risk patients who accounted for 45.2% (43 out of 95) of this subgroup, meaning that, from POD4 onwards, we could be highly alert for anastomosis leakage mainly for the remaining 52 patients with a cut-off ≥ 6.5–21.2% of the initial sample (52 out of 245). 

However, we should underline that NLRs failed to identify five leaks out of twenty-eight (rate 17.85%), one case on POD1 and four on POD4, of which one was of Grade C, reoperated, two of Grade B, and two of Grade C—totally asymptomatic. At the same time, we should emphasize that, in fact, we had to be alert for 21.2% of the initial sample of patients (52 out of 245). This, practically, means that, in the era of laparoscopic and Fast Tract surgery, most patients may be discharged earlier, without the fear of an anastomotic rupture after early discharge, and only 21.2% of patients in our material needed to remain hospitalized, being of high risk for leakage.

Our findings are in agreement with those of Paliogiannis [4], who also reported positive predictive findings on both POD1 and POD4, with the best prediction of a leakage on POD4 (cut-off ≥ 7.1). Furthermore, Mik et al. [7], in 724 patients, and Walker, in only 136 patients, support the NLR value on POD4, with a cut-off value ≥ 6.5 and 6.15, respectively. Regarding our finding of NLR prediction on POD1 with a cut-off ≥ 7.4 with 68.7% sensitivity and 96.4% specificity, and a NPV of 99.3, we are in disagreement with Cook [40], who found a cut-off ≥ 9.3 with 66% sensitivity and 69% specificity.

In our opinion, the most important finding of our study comes from the analysis of the components of NLR; that is, neutrophils and lymphocytes. As we can see in Table 3, which presents lymphocyte counts, patients who finally experienced a leak were found to have severe lymphopenia on POD3, POD4, and POD5, in relation to the non-L group. There are no similar observations/comments in other publications, but it is absolutely known that, for postoperative lymphopenia, reaching its nadir from between two hours to two days after surgery is a common finding. Since lymphocytes are a major component of infection control, postoperative lymphopenia is considered a risk factor for postoperative infection complications [6,27], while Takahashi et al. [49] reported data suggesting that lymphopenia (under 10% or 1.0 × 10^9^ cells/L) for 4 days postoperatively is predictive of surgical wound infection. Of course, in cancer patients, and particularly those with progressive disease, T lymphocyte homeostasis is highly impaired, leading to a loss of appropriate T lymphocyte responses [50].

However, there are some limitations to our study: First, it is a retrospective study, with only a moderate number or participants. Only half of our population had been operated on for low or very low anterior resection. This sounds to be positive, in comparison with other studies with very different types of colectomies; however, this number (n = 123) alone is insufficient to draw conclusions as to the NLR’s predictive value, which would be ideal. Second, it is a single center study, which of course has the inherent limitation of the results not being generalizable. On the other hand, the “single center” collected patients ensure the maximum homogeneity of the “material” since, in our cases, all the patients were operated on by the same colorectal surgeon and treated by the same surgical group. Third, preventive colostomies were performed at the discretion of the surgeon—the same chief surgeon in every case—and not as a routine. This practice could indirectly affect the healing of an anastomosis, leading to a leak. Finally, we group together the 18 cases of clinically recognized leaks and another 10 cases, which were eventually seen at endoscopy 3 months postoperatively. It should be mentioned that these 10 cases (Grade C), although totally asymptomatic, exhibit the same characteristics in respect to the neutrophil and lymphocyte values—in other words, they had the same risk for leak. 

## 5. Conclusions

The present study in 245 patients, 123 with rectal and 122 with colon cancer, led us to suggest that as early as POD1 there is a highly significant difference in the NLR value between potential non-leak and leak groups. Patients having a cut-off value ≥ 7.4 enable a negative prediction for leakage, with an AUC of 0.881, a sensitivity of 68.7%, a specificity of 96.4%, a PPV of 28.4%, and a NPV of 99.3%. This finding gives us the advantage of being able to exclude 150 patients from the suspicion of leak (low probability of risk). Additionally, on POD4, patients with a cut-off point ≥ 6.5 enable a positive prediction for leakage, with an AUC of 0.698, a sensitivity of 82.1%, a specificity of 51.6%, a PPV of 17.6%, and a NPV of 95.6%. This also means that patients with a cut-off point lower than 6.5 on POD4 are at risk, but a much lower risk (43 patients, of whom only four finally presented leakage). This finding further enables us to identify low-risk patients, leaving only 52 out of the total 245 (rate 21.2%) as candidates for anastomotic leakage.

From the above findings, we suggest the NLR score as having a highly significant ability to accurately predict the likelihood of anastomotic leakage, as early as POD1. This makes our study significantly innovative. However, further studies with more homogenic populations should give even more accurate results.

## Data Availability

The data and materials/figures used in the current study are available from the corresponding author on reasonable request.

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
