# Peer review of "Negative and Positive Predictors of Anastomotic Leakage in Colorectal Cancer Patients—The Case of Neutrophil-to-Lymphocyte Ratio"

_diagnostics, 2024, doi:10.3390/diagnostics14161806_

Round 1

Reviewer 1 Report

Comments and Suggestions for Authors

The primary research question explores the predictive value of the neutrophil-to-lymphocyte ratio (NLR) for anastomotic leakage in colorectal cancer surgery patients. The study aims to identify the most reliable postoperative day for predicting leakage risk using NLR values.

Anastomotic leakage is a severe complication, and early prediction could significantly improve patient outcomes through personalized treatment strategies. Thus the topic is of high clinical importance/relevance. 

The manuscript is well-structured and writtne. 

The methodology is robust, involving a comprehensive evaluation of NLR values from the day before surgery up to POD7 in a cohort of 245 patients. 

In the results section, the identification of specific cut-off points for NLR on POD1 and POD4 provides actionable insights for clinical practice. The study's findings are supported by statistical analysis.

The discussion thoroughly examines the implications of the findings, comparing them with existing studies on NLR and anastomotic leakage. The authors acknowledge discrepancies in the literature regarding the most predictive postoperative day and offer plausible explanations for these differences. The analysis of NLR components (neutrophils and lymphocytes) and their association with postoperative stress response is also reported. It would also be useful to insert 1-2 parragraphs related to the importance of personalized therapeutic approaches in achieving significant downstaging and clinical remission, which can be compared with the predictive strategies discussed in your study (e.g. Pereira C, Mohan J, Gururaj S, Chandrashekhara P. Predictive Ability of Neutrophil-Lymphocyte Ratio in Determining Tumor Staging in Colorectal Cancer. Cureus. 2021;13(10):e19025. Published 2021 Oct 25. doi:10.7759/cureus.19025; Cui, M., Xu, R., & Yan, B. (2020). A persistent high neutrophil‑to‑lymphocyte ratio predicts poor prognosis in patients with colorectal cancer undergoing resection. Molecular and Clinical Oncology, 13, 63. https://doi.org/10.3892/mco.2020.2133; Georgescu, D. E., Georgescu, M. T., Bobircă, F. T., Georgescu, T. F., Doran, H., & Pătraşcu, T. (2017). Synchronous Locally Advanced Rectal Cancer with Clinical Complete Remission and Important Downstaging after Neoadjuvant Radiochemotherapy - Personalised Therapeutic Approach. Chirurgia (Bucharest, Romania : 1990)112(6), 726–733. https://doi.org/10.21614/chirurgia.112.6.726). 

The conclusions identify   the potential of NLR on POD1 to identify low-risk patients and on POD4 to identify high-risk patients for anastomotic leakage.  This dual predictive value could guide clinicians in tailoring postoperative care and monitoring more effectively.

The study has some limitations: retrospective type, moderate sample size, and potential variability in patient populations and surgical practices. 

Author Response

REVIEWER 1

corrections in the text are highlighted in green color

The primary research question explores the predictive value of the neutrophil-to-lymphocyte ratio (NLR) for anastomotic leakage in colorectal cancer surgery patients. The study aims to identify the most reliable postoperative day for predicting leakage risk using NLR values. Anastomotic leakage is a severe complication, and early prediction could significantly improve patient outcomes through personalized treatment strategies. Thus, the topic is of high clinical importance/relevance. 

Q1.    The manuscript is well-structured and written. The methodology is robust, involving a comprehensive evaluation of NLR values from the day before surgery up to POD7 in a cohort of 245 patients. In the results section, the identification of specific cut-off points for NLR on POD1 and POD4 provides actionable insights for clinical practice. The study's findings are supported by statistical analysis. The discussion thoroughly examines the implications of the findings, comparing them with existing studies on NLR and anastomotic leakage. The authors acknowledge discrepancies in the literature regarding the most predictive postoperative day and offer plausible explanations for these differences. The analysis of NLR components (neutrophils and lymphocytes) and their association with postoperative stress response is also reported.

I appreciate your positive comments

Q2.    It would also be useful to insert 1-2 paragraphs related to the importance of personalized therapeutic approaches in achieving significant downstaging and clinical remission, which can be compared with the predictive strategies discussed in your study (e.g. Pereira C, Mohan J, Gururaj S, Chandrashekhara P. Predictive Ability of Neutrophil-Lymphocyte Ratio in Determining Tumor Staging in Colorectal Cancer. Cureus. 2021;13(10):e19025. Published 2021 Oct 25. doi:10.7759/cureus.19025;        Cui, M., Xu, R., & Yan, B. (2020). A persistent high neutrophil‑to‑lymphocyte ratio predicts poor prognosis in patients with colorectal cancer undergoing resection. Molecular and Clinical Oncology, 13, 63. https://doi.org/10.3892/mco.2020.2133;     Georgescu, D. E., Georgescu, M. T., Bobircă, F. T., Georgescu, T. F., Doran, H., & Pătraşcu, T. (2017). Synchronous Locally Advanced Rectal Cancer with Clinical Complete Remission and Important Downstaging after Neoadjuvant Radiochemotherapy - Personalised Therapeutic Approach. Chirurgia (Bucharest, Romania : 1990)112(6), 726–733. https://doi.org/10.21614/chirurgia.112.6.726). 

Thank you for your suggestion. Although our study is exclusively focused on the research of the best predictive NLR value as a marker of anastomotic leakage and not on the assessment of the usefulness of NLR as a predictive index, we have added two of the references you have suggested – the last one is a case-report not related to NLR - along with a comment on them, please see lines 55-56, where we have added “Additionally, it has been used as an indicator of colorectal tumor staging [Pereira C, Mohan J, Gururaj S, Chandrashekhara P. Predictive Ability of Neutrophil-Lymphocyte Ratio in Determining Tumor Staging in Colorectal Cancer. Cureus. 2021;13(10):e19025. Published 2021 Oct 25. doi:10.7759/cureus.19025] as well as a predictor of poor, long-term postoperative prognosis [Cui, M., Xu, R., & Yan, B. (2020). A persistent high neutrophil‑to‑lymphocyte ratio predicts poor prognosis in patients with colorectal cancer undergoing resection. Molecular and Clinical Oncology, 13, 63. https://doi.org/10.3892/mco.2020.2133].”

Q3.    The conclusions identify   the potential of NLR on POD1 to identify low-risk patients and on POD4 to identify high-risk patients for anastomotic leakage.  This dual predictive value could guide clinicians in tailoring postoperative care and monitoring more effectively.

Thank you.

Q4.    The study has some limitations: retrospective type, moderate sample size, and potential variability in patient populations and surgical practices. 

We have recognized these limitations – please see Discussion, last paragraph

Reviewer 2 Report

Comments and Suggestions for Authors

This study provides valuable insights into the prediction of postoperative complications in colorectal cancer, particularly in identifying high-risk patients through the early postoperative Neutrophil-to-Lymphocyte Ratio (NLR). However, there are several areas that require further clarification and improvement:

  1. The data for this study were collected from a single center, which may limit the generalizability of the results. Additionally, the sample size included in the study is relatively small, with only a few cases of anastomotic leakage (10 cases), which may affect the robustness of the statistical results and the reliability of the conclusions.
  2. The article primarily focuses on the statistical predictive ability of NLR but lacks a deeper explanation of the physiological mechanisms behind NLR changes during the occurrence of anastomotic leakage. While the article mentions the relationship between NLR and the inflammatory response, it does not thoroughly explore why significant changes in NLR occur at specific postoperative time points and how these changes are biologically linked to postoperative complications. A more in-depth discussion of the physiological mechanisms would enhance the clinical credibility and applicability of the study's findings.
  3. The data presentation in the article relies mainly on basic statistical analysis and does not fully utilize data visualization tools to strengthen the presentation and interpretation of the results. For instance, the dynamic changes in NLR over time could be more effectively communicated using more intuitive graphical representations to clearly convey its predictive value. Additionally, the statistical analysis in the article is mostly focused on sensitivity and specificity, but lacks further exploration and detailed analysis of the data, such as subgroup analysis or multivariate regression models, which could provide richer insights.

Author Response

REVIEWER 2

corrections in the text are highlighted in yellow color

This study provides valuable insights into the prediction of postoperative complications in colorectal cancer, particularly in identifying high-risk patients through the early postoperative Neutrophil-to-Lymphocyte Ratio (NLR). However, there are several areas that require further clarification and improvement:

Q1.    The data for this study were collected from a single center, which may limit the generalizability of the results. Additionally, the sample size included in the study is relatively small, with only a few cases of anastomotic leakage (10 cases), which may affect the robustness of the statistical results and the reliability of the conclusions.

We recognize the disadvantage of the “single center” and so we have inserted this sentence in study limitations: please see lines 411 -414.           “Second, it is a single center study, which of course has the inherent limitation of the results not being generalizable. On the other hand, the “single center” collected patients ensure the maximum homogeneity of the “material”, since, in our cases all the patients were operated on by the same colorectal surgeon and treated by the same surgical group”.

Regarding the sample size [245 cases], we consider that it is “moderate” and not small; and we have already commented on this in study limitations – line 407.

Regarding the cases of anastomotic leakage, there are 28 and not 10 cases, that is 11.4% of the total operated patients, please see line 162, Table 1.

Q2.    The article primarily focuses on the statistical predictive ability of NLR but lacks a deeper explanation of the physiological mechanisms behind NLR changes during the occurrence of anastomotic leakage. While the article mentions the relationship between NLR and the inflammatory response, it does not thoroughly explore why significant changes in NLR occur at specific postoperative time points and how these changes are biologically linked to postoperative complications. A more in-depth discussion of the physiological mechanisms would enhance the clinical credibility and applicability of the study's findings.

Thank you, it is an interesting point. Unfortunately, our focus was “to assess the best predictive value as a marker of anastomotic leakage” [lines 71-72] and not to try to explain in depth the pathophysiology of NLR changes during the anastomotic leakage process, this knowledge being – more or less – unexplored as yet. However, in the discussion [lines 394 – 404], we have already made some comments, suggesting a [theoretical] connection between anastomotic leakage and inflammation [neutrophils] as well as immune function [lymphocytes], mainly based on our previous work on NLR in ventilatory-associated pneumonia in polytrauma patients who did or did not receive probiotics “Menni AE, et al. The Effect of Probiotics on the Prognostication of the Neutrophil-to-Lymphocyte Ratio in Severe Multi-Trauma Patients. J Pers Med. 2024 Apr 15;14(4):419. doi: 10.3390/jpm14040419.”. We have added this reference in line 401.

Q3.    The data presentation in the article relies mainly on basic statistical analysis and does not fully utilize data visualization tools to strengthen the presentation and interpretation of the results. For instance, the dynamic changes in NLR over time could be more effectively communicated using more intuitive graphical representations to clearly convey its predictive value. Additionally, the statistical analysis in the article is mostly focused on sensitivity and specificity, but lacks further exploration and detailed analysis of the data, such as subgroup analysis or multivariate regression models, which could provide richer insights.

Thank you for your comments. Regarding your 1st comment, we apologize, but there are no other tools in the statistical package we have used [SPSS, v25.0], nor in the Microsoft Excel. Your suggestion of an alternative tool would be valuable.

Regarding the 2nd comment, we really focused on sensitivity and specificity, as all other authors dealing with predictive value of NLR [in whatever disease or clinical situation] do. Of course, a subgroup analysis of our data would be of interest. However no significant results were found after such analysis; the same is true and after analysis of only 123 patients – 4 leaks, or 122 patients -24 leaks [low anterior resection patients], which we have commented on in the limitations of our study [lines 409 - 410].

Reviewer 3 Report

Comments and Suggestions for Authors

The article focuses on the role of the Neutrophil-to-Lymphocyte Ratio (NLR) as a predictive marker for anastomotic leakage in colorectal cancer patients following surgery. The study analyzes NLR values at different postoperative days to determine their effectiveness in predicting leakage, which is a serious complication in colorectal surgeries. The findings suggest that NLR on postoperative day 1 (POD1) and day 4 (POD4) are particularly indicative of patients' risk levels for developing an anastomotic leak, with higher NLR values associated with higher risk. The study concludes that NLR could be a valuable tool for early identification of patients at risk, potentially improving outcomes by enabling timely interventions.

I recommend that colon and rectal surgeries be analyzed separately, as they present different risks for anastomotic leakage. It would be more precise to focus on one type of surgery for your analysis. Additionally, for rectal cancer patients, it's important to account for neoadjuvant treatments such as chemoradiotherapy, which significantly increase the risk of leakage. I also suggest incorporating other variables that could influence the leak risk, such as smoking, vascular disease, diabetes, and other comorbidities.

Additionally, it would be valuable to include intraoperative and postoperative data, such as the rate and type of complications and the duration of surgery, to determine whether these factors are also associated with an increased risk of anastomotic leakage.

Please also include a flowchart detailing the patient selection process.

I find the work compelling, but it would benefit from those adjustments.

Author Response

REVIEWER 3

corrections in the text are highlighted in turquoise color

The article focuses on the role of the Neutrophil-to-Lymphocyte Ratio (NLR) as a predictive marker for anastomotic leakage in colorectal cancer patients following surgery. The study analyzes NLR values at different postoperative days to determine their effectiveness in predicting leakage, which is a serious complication in colorectal surgeries. The findings suggest that NLR on postoperative day 1 (POD1) and day 4 (POD4) are particularly indicative of patients' risk levels for developing an anastomotic leak, with higher NLR values associated with higher risk. The study concludes that NLR could be a valuable tool for early identification of patients at risk, potentially improving outcomes by enabling timely interventions.

Q1.    I recommend that colon and rectal surgeries be analyzed separately, as they present different risks for anastomotic leakage. It would be more precise to focus on one type of surgery for your analysis.

Thank you for your suggestion. Of course, a subgroup analysis of our data would be of interest. However no significant results were found after such analysis; the same, no significant results, were found after analysis of only one group [123 patients – 4 leaks, OR 122 patients -24 leaks - low anterior resection patients]; unfortunately, there are no significantly different findings and no prediction. We have already commented this in the limitations of our study [lines 409 – 410].

Q2. Additionally, for rectal cancer patients, it's important to account for neoadjuvant treatments such as chemoradiotherapy, which significantly increase the risk of leakage. I also suggest incorporating other variables that could influence the leak risk, such as smoking, vascular disease, diabetes, and other comorbidities.

It is an interesting point. All low anterior resection subjected patients had been previously subjected to chemoradiotherapy, and are therefore at the same risk of experiencing an anastomosis leakage. Regarding the other causative factors involved in leakage pathology, unfortunately, we have no access to such data – it is a retrospective study – and most importantly, we had only a moderate number of patients to deal with. By splitting them into categories of theoretical risk factors for leakage it seems impossible to draw any conclusion. And, on the other hand, the only purpose of our study was “to assess the best predictive value as a marker of anastomotic leakage” [lines 71 -72] of whatever underlying, causative factor.

Q3.    Additionally, it would be valuable to include intraoperative and postoperative data, such as the rate and type of complications and the duration of surgery, to determine whether these factors are also associated with an increased risk of anastomotic leakage.

Unfortunately, we have no access to data related to additional complications. However, we dare to assume that, since the curve of neutrophil values shows only a moderate increase up to POD2 or POD3 and then a smooth, gradual decrease until the POD7, there are no apparent complications up to POD7 [Figure 1].

Regarding the duration of surgery, this was more or less the same in all patients, since all operations were performed by the same colorectal surgeon and laparoscopic procedures were excluded from analysis [lines 84 and 87 - 88].

Q4.       Please also include a flowchart detailing the patient selection process.

OK, please find it as FlowChart [line 85].

Q5.       I find the work compelling, but it would benefit from those adjustments.

Thank you!

Round 2

Reviewer 2 Report

Comments and Suggestions for Authors

The authors' replies to the reviewers' comments reflect an awareness of the study's limitations and have resolved some of the issues raised by the reviewers to a certain degree. However, for the suggestions on data visualization and in-depth statistical analysis, the authors have not provided any improvement measures, which could be an area for consideration in future research endeavors. Overall, this is a highly significant work, and it is recommended for acceptance and publication.